# Evidence for a sexual dimorphism in gene expression noise in metazoan species

Carlos Díaz-Castillo

Department of Developmental & Cell Biology, University of California, Irvine, CA, USA

## ABSTRACT

Many biological processes depend on very few copies of intervening elements, which makes such processes particularly susceptible to the stochastic fluctuations of these elements. The intrinsic stochasticity of certain processes is propagated across biological levels, causing genotype- and environment-independent biological variation which might permit populations to better cope with variable environments. Biological variations of stochastic nature might also allow the accumulation of variations at the genetic level that are hidden from natural selection, which might have a great potential for population diversification. The study of any mechanism that resulted in the modulation of stochastic variation is, therefore, of potentially wide interest. I propose that sex might be an important modulator of the stochastic variation in gene expression, i.e., gene expression noise. Based on known associations between different patterns of gene expression variation, I hypothesize that in metazoans the gene expression noise might be generally larger in heterogametic than in homogametic individuals. I directly tested this hypothesis by comparing putative genotype- and environment-independent variations in gene expression between females and males of *Drosophila melanogaster* strains. Also, considering the potential effect of the propagation of gene expression noise across biological levels, I indirectly tested the existence of a metazoan sexual dimorphism in gene expression noise by analyzing putative genotype- and environment-independent variation in phenotypes related to interaction with the environment in *D. melanogaster* strains and metazoan species. The results of these analyses are consistent with the hypothesis that gene expression is generally noisier in heterogametic than in homogametic individuals. Further analyses and discussion of existing literature permits the speculation that the sexual dimorphism in gene expression noise is ultimately based on the nuclear dynamics in gametogenesis and very early embryogenesis of sex-specific chromosomes, i.e., $Y$ and $W$ chromosomes.

Corresponding author
Carlos Díaz-Castillo,
crlsdiazcastillo@gmail.com,
cdiazcas@uci.edu

## INTRODUCTION

Biological systems are prone to vary even in the absence of genetic modifications or environmental changes (*Burga & Lehner, 2012*; *Feinberg & Irizarry, 2010*; *Kaern et al., 2005*; *Kilfoil, Lasko & Abouheif, 2009*; *Lehner, 2013*; *Raj & Van Oudenaarden, 2008*;

*Raser & O'Shea, 2005*). Although not yet completely understood, biological variation in the absence of genetic or environmental cues ultimately depends on stochastic transitions and the interaction of elements that contribute to biological processes (*Burga & Lehner, 2012*; *Feinberg & Irizarry, 2010*; *Kaern et al., 2005*; *Kilfoil, Lasko & Abouheif, 2009*; *Lehner, 2013*; *Raj & Van Oudenaarden, 2008*; *Raser & O'Shea, 2005*). One of the biological processes known to vary even in the absence of genetic or environmental cues is gene expression (*Burga & Lehner, 2012*; *Feinberg & Irizarry, 2010*; *Kaern et al., 2005*; *Kilfoil, Lasko & Abouheif, 2009*; *Lehner, 2013*; *Raj & Van Oudenaarden, 2008*; *Raser & O'Shea, 2005*). The multiple steps encompassed by gene expression depend on a very small amount of intervening elements, making gene expression particularly susceptible to the intrinsic stochasticity of these elements, transitions and interactions (*Burga & Lehner, 2012*; *Feinberg & Irizarry, 2010*; *Kaern et al., 2005*; *Kilfoil, Lasko & Abouheif, 2009*; *Lehner, 2013*; *Raj & Van Oudenaarden, 2008*; *Raser & O'Shea, 2005*).

Biological variation ultimately caused by stochastic events at the molecular level can be an important evolutionary driving force (*Burga & Lehner, 2012*; *Feinberg & Irizarry, 2010*; *Kaern et al., 2005*; *Kilfoil, Lasko & Abouheif, 2009*; *Lehner, 2013*; *Raj & Van Oudenaarden, 2008*; *Raser & O'Shea, 2005*). On one side, stochastic biological variation represents a cheap phenotypic diversification that might permit populations to cope with variable environments (*Burga & Lehner, 2012*; *Feinberg & Irizarry, 2010*; *Kaern et al., 2005*; *Kilfoil, Lasko & Abouheif, 2009*; *Lehner, 2013*; *Raj & Van Oudenaarden, 2008*; *Raser & O'Shea, 2005*). On the other side, stochastic biological variation might act as a genetic capacitor (*Chalancon et al., 2012*). Genetic capacitance refers to the accumulation of genetic variation with no phenotypic effect, i.e., cryptic genetic variation and its release upon capacitance attenuation (*Masel, 2013*; *Masel & Trotter, 2010*; *Paaby & Rockman, 2014*). Genetic variation with phenotypes indistinguishable from the spectrum of stochastic phenotypes will be allowed to accumulate in a cryptic state until stochastic capacitance is somehow attenuated and cryptic genetic variation becomes phenotypically relevant (*Chalancon et al., 2012*; *Paaby & Rockman, 2014*). Thus, stochastic biological variation could provide natural populations with short-term endurance to environmental variables, while permitting the accumulation of cryptic genetic variation with great potential for diversification. Although stochastic variation is an intrinsic property of biological systems, any factor that modulated stochastic biological variation would have direct and capacitance-driven indirect effects on short-term responses to environmental changes and the long-term diversification of natural populations. Therefore, the identification of factors that resulted in the modulation of stochastic variation is of broad interest.

The integration of literature on gene expression variation suggests sex can be an important modulator of stochastic biological variation in metazoan species. On one hand, as would be expected from the contribution of any source of stochastic biological variation to genetic capacitance, stochastic variation in gene expression, or gene expression noise, has been shown to correlate positively with gene expression variation in response to conditional changes and divergence (*Dong et al., 2011*). Genetic variation for loci with a noisier gene expression would be prone to accumulate in a cryptic state until capacitance

is attenuated, whereas genetic variation for loci with a less noisy gene expression would be more often phenotypically noticeable, and removed from populations if detrimental. On the other hand, male-biased expression in dipterans and mammals responds to conditional changes and diverges faster than homogametic female-biased or unbiased gene expression, whereas in birds it is the female-biased expression that has a faster conditional response and divergence (*Assis, Zhou & Bachtrog, 2012*; *Ellegren & Parsch, 2007*; *Gallach, Domingues & Betran, 2011*; *Jiang et al., 2010*; *Mank, 2009*; *Mank et al., 2007*; *Meisel, 2011*; *Parsch & Ellegren, 2013*; *Singh & Artieri, 2011*; *Wyman, Agrawal & Rowe, 2010*; *Wyman, Cutter & Rowe, 2011*). In dipterans and mammals, females are homogametic and males are heterogametic; in birds, females are heterogametic and males homogametic. Thus, it could be generalized that heterogametic sex-biased gene expression in metazoan species responds to conditional changes and diverges faster than homogametic sex-biased or unbiased gene expression (*Assis, Zhou & Bachtrog, 2012*; *Ellegren & Parsch, 2007*; *Gallach, Domingues & Betran, 2011*; *Jiang et al., 2010*; *Mank, 2009*; *Mank et al., 2007*; *Meisel, 2011*; *Parsch & Ellegren, 2013*; *Singh & Artieri, 2011*; *Wyman, Agrawal & Rowe, 2010*; *Wyman, Cutter & Rowe, 2011*).

Considering the contribution of gene expression noise to genetic capacitance, the differences in sex-biased gene expression variation noticed in metazoan species could be explained if gene expression was generally noisier in the heterogametic sex (Fig. 1). Phenotypes resulting from genetic variation in loci that are expressed in heterogametic individuals would often be indistinguishable from the broad spectrum of phenotypes dependent on the generally noisier gene expression of the heterogametic sex. Phenotypes resulting from genetic variation in loci that are expressed in homogametic individuals would often be distinguishable from the narrower spectrum of phenotypes dependent on the less noisy gene expression of the homogametic sex, and could be purged from populations if detrimental. Therefore, genetic variation for loci that are specifically expressed or overexpressed in the heterogametic sex would be prone to accumulate in a cryptic state, and become phenotypically relevant when capacitance is somehow attenuated by conditional changes in single and/or divergent populations.

In this article, I present evidence for the existence of a sexual dimorphism in gene expression noise in metazoan species by comparing female and male putative genotype- and environment-independent variation for transcript abundance and other phenotypic traits in *Drosophila melanogaster*, and by analyzing sex-biased dispersal in metazoan species. These analyses are consistent with the hypothesis that gene expression is generally noisier in the heterogametic sex, and point to the possibility that this sexual dimorphism in gene expression noise might be ultimately dependent on sex-specific chromosomes, i.e., *Y* and *W*. Mechanistic details of the effect of sex-specific chromosomes on gene expression noise can be found in the 'Results and Discussion' section.

## MATERIALS AND METHODS

The original sources of the datasets used in this article are: *D. melanogaster* transcript abundance in adult females and males (*Diaz-Castillo, Xia & Ranz, 2012*), the DGRP strains phenotypic data (*Huang et al., 2014*; *Mackay et al., 2012*), coordinates of regions
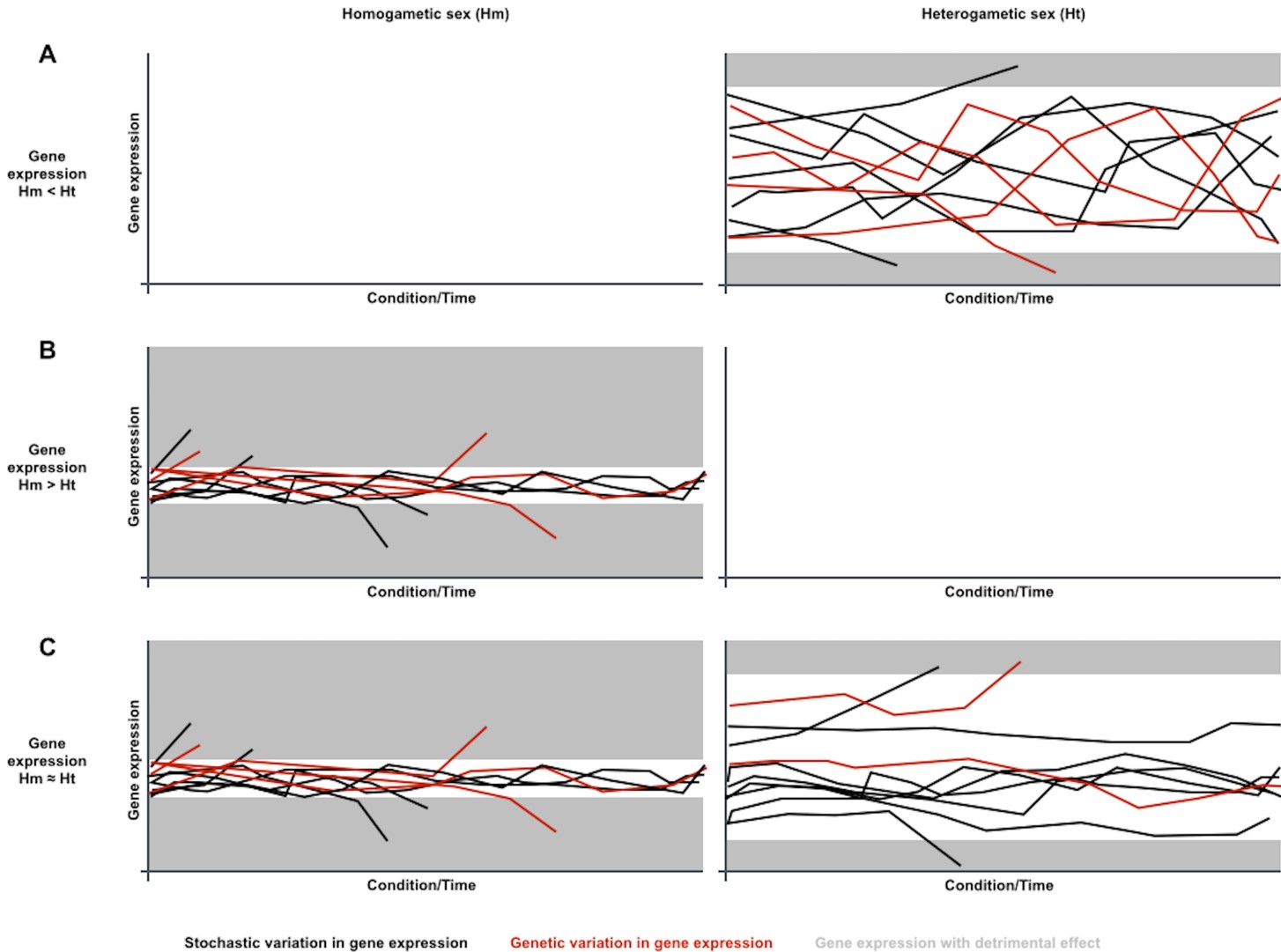

**Figure 1** **A sexual dimorphism in gene expression noise could explain differences in conditional response and divergence for sex-biased gene expression.** Charts symbolize gene expression dynamics for three transcripts in a population under different conditions or along time, under the assumption that gene expression is generally noisier in the heterogametic sex than in the homogametic sex: (A) Dynamics for a transcript that is overexpressed in the generally noisier heterogametic sex ($Hm < Ht$). (B) Dynamics for a transcript that is overexpressed in the generally less noisy homogametic sex ($Hm > Ht$). (C) Dynamics for a transcript equally expressed in both sexes ($Hm \approx Ht$). Each line represents transcript abundance variation for a single individual in the population. Black and red lines represent variation in gene expression of stochastic or genetic nature, respectively. Grey areas represent gene expression levels with detrimental effects. In this case, both overexpression and underexpression beyond certain levels are detrimental. Individuals with detrimental expression stop contributing to the population. Noisier gene expression increases endurance for environmental changes, symbolized with narrower ranges for detrimental expression. Genetic variation in gene expression is prone to accumulate for transcripts that are overexpressed in the noisiest sex, as their phenotypes are often indistinguishable from the noise-driven phenotypic spectrum (A). Genetic variation in gene expression is less prone to accumulate for transcripts that are overexpressed in the less noisy sex or equally expressed in both sexes, as their phenotypes are often distinguishable from the noise-driven phenotypic spectrum, and removed from the population if detrimental (B and C).

with distinctive chromatin features in *D. melanogaster* somatic cells (*Filion et al., 2010*), and sex-biased dispersal in metazoan species (*Petit & Excoffier, 2009*). Coordinates of genes coding for transcripts in the dataset of Diaz-Castillo, Xia and Ranz and associates were obtained from FlyBase (*Diaz-Castillo, Xia & Ranz, 2012*; *St Pierre et al., 2014*). Microsoft® Excel® for Mac 2011 (Microsoft Corporation) was used to perform Monte Carlo–Wilcoxon matched-pairs signed-ranks tests, and Monte Carlo simulations. Prism 5 for Mac OS X (GraphPad Software, Inc) was used to analyze polynomial regressions between transcript abundance mean and CV for *D. melanogaster* strains and genotypes (*Diaz-Castillo, Xia & Ranz, 2012*). Data processing and statistical analyses are described in detail in the 'Results and Discussion' section and in the Table footnotes.

## RESULTS AND DISCUSSION

### Direct evidence for the existence of a sexual dimorphism in gene expression noise in *Drosophila*

The existence of the hypothesized sexual dimorphism in gene expression noise should be easy to test. When directly comparing gene expression between sexes, putative genotype- and environment-independent variation should be generally larger in heterogametic than in homogametic individuals. *Diaz-Castillo, Xia & Ranz (2012)* performed genome-wide microarray-based transcript abundance analyses of adult naive females and males in six *D. melanogaster* strains. These analyses resulted in abundance data of 16,637 transcripts for three biological replicates per strain and sex (36 samples: 6 strains × 2 sexes × 3 biological replicates). All analyzed samples were maintained under the same environmental conditions, and RNA was extracted simultaneously to minimize environmental and technical variation between biological replicates. Genotype and environment were found to be identical for biological replicates of each strain, with the only exception being of the dosage of sex chromosomes, i.e., *X* and *Y*. Thus, any transcript abundance variation detected for biological replicates of a single strain might only be of sexual and/or stochastic nature, making it possible to test the potential existence of a generalized difference in gene expression noise between sexes.

Interestingly, five of the analyzed strains shared the same exact genotype with at least one another strain. INV1 and INV2 genotype is $w^{1118}/y^+$ $Y$; $In(2R)[P\{FRT\}^{CB\text{-}0236\text{-}3}, P\{FRT, w^+\}^{5\text{-}HA\text{-}1995}]/SM6a$, whereas SIM1, REV1 and REV2 genotype is $w^{1118}/y^+$ $Y$; $P\{FRT, w^-\}^{CB\text{-}0236\text{-}3}, P\{FRT, w^-\}^{5\text{-}HA\text{-}1995}/SM6a$. Since all samples were reared in the same conditions, the transcript abundance data of strains with the same genotype could be pooled together to perform analyses with increasing analytical power (6 and 9 biological replicates per sex for INV and SIM/REV genotypes, respectively).

Since females in *Drosophila* are homogametic and males are heterogametic, it would be expected that, in the dataset under study, measures for putative stochastic variation in transcript abundance were generally larger in males than in females. To test this prediction, I retrieved *Diaz-Castillo, Xia & Ranz*'s (*2012*) dataset from the Gene Expression Omnibus database (GSE31120). Transcript abundance data in this dataset had been normalized to permit comparisons between samples (*Diaz-Castillo, Xia & Ranz, 2012*). To compare

**Table 1  Monte Carlo–Wilcoxon matched-pairs signed-ranks tests for putative stochastic variation in transcript abundance in *D. melanogaster* females and males.**

| Strains | N replicates | N data pairs | Observed W | Simulated W [5th/95th percentiles] | $P_{upper}/P_{lower}$ |
|---------|--------------|--------------|------------|-----------------------------------|----------------------|
| REC | 3 | 16,637 | $-1.21 \times 10^8$ | $-2.06 \times 10^6/2.07 \times 10^6$ | 1.0000/<0.0001 |
| INV1 | 3 | 16,637 | $-1.24 \times 10^8$ | $-2.04 \times 10^6/2.03 \times 10^6$ | 1.0000/<0.0001 |
| INV2 | 3 | 16,637 | $-1.88 \times 10^6$ | $-2.03 \times 10^6/2.04 \times 10^6$ | 0.9354/0.0646 |
| SIM1 | 3 | 16,637 | $-1.09 \times 10^8$ | $-1.97 \times 10^6/2.05 \times 10^6$ | 1.0000/<0.0001 |
| REV1 | 3 | 16,637 | $-5.96 \times 10^7$ | $-2.05 \times 10^6/2.08 \times 10^6$ | 1.0000/<0.0001 |
| REV2 | 3 | 16,637 | $-1.49 \times 10^7$ | $-2.03 \times 10^6/2.02 \times 10^6$ | 1.0000/<0.0001 |
| INV | 6 | 16,637 | $-1.18 \times 10^8$ | $-2.00 \times 10^6/2.03 \times 10^6$ | 1.0000/<0.0001 |
| SIM/REV | 9 | 16,637 | $-1.05 \times 10^8$ | $-2.00 \times 10^6/2.04 \times 10^6$ | 1.0000/<0.0001 |

**Notes.**

Transcript abundance data was obtained from *Diaz-Castillo, Xia & Ranz (2012)*. Observed Wilcoxon W were obtained by subtracting $CV_M$ from $CV_F$ for each element in the dataset for each strain or genotype. CV differences were ranked from lower to higher according to their absolute value. Signs were assigned to ranks according to the sign of CV differences. W were obtained by adding signed ranks for all the elements in the dataset for each strain or genotype. Simulated W were obtained by repeating the same process after randomly rearranging all CV values for each strain or genotype 10,000 times. $P_{upper}$ and $P_{lower}$ values represent the fraction of random simulations with measures larger or equal, and lower or equal, than the observed ones, respectively.

transcript abundance variation putatively independent of genetic and environmental cues, I calculated transcript abundance coefficients of the variation for females and males in each strain and after pooling data of strains with the same genotype ($CV_F$ and $CV_M$, respectively). The existence of a general trend for the difference of putative stochastic variation in transcript abundance between sexes was tested by performing Monte Carlo–Wilcoxon matched-pairs signed-ranks tests for each strain and genotype. For each transcript in the dataset, I subtracted $CV_M$ from $CV_F$. CV differences were ranked upon the absolute value of their difference from lower to higher, and signs were assigned to each rank upon the sign of the difference between $CV_F$ and $CV_M$. Wilcoxon sums of signed ranks (W) were calculated by adding signed ranks for all the transcripts in the dataset and for each strain and genotype. W is perfectly suited for the identification of general biases in collections of paired data because it is sensitive both to the number of elements in the dataset with biased measures and the extent of such biases, thus eliminating the need to set arbitrary thresholds to infer trends of putative biologically significance. To estimate the significance of observed W, I recalculated W after randomly rearranging all CV data for each strain and genotype 10,000 times. Simulated W would represent the value W can adopt for a dataset with the same number of elements and the same value range without existing significant differences between sexes.

W ultimately depends on the subtraction of $CV_M$ from $CV_F$ for each transcript. If gene expression was indeed generally noisier in males than in females, it would be expected that observed W were negative and commonly lower than simulated ones. In fact, with the exception of strain INV2, all observed W were negative and lower than any simulated W (Table 1 and Fig. 2). Strains INV1 and INV2 are genotypically identical, yet differ with regard to the significance of the hypothesized sexual dimorphism in gene expression noise. It is worth noting that observed W are considerably more variable for single-strain analyses

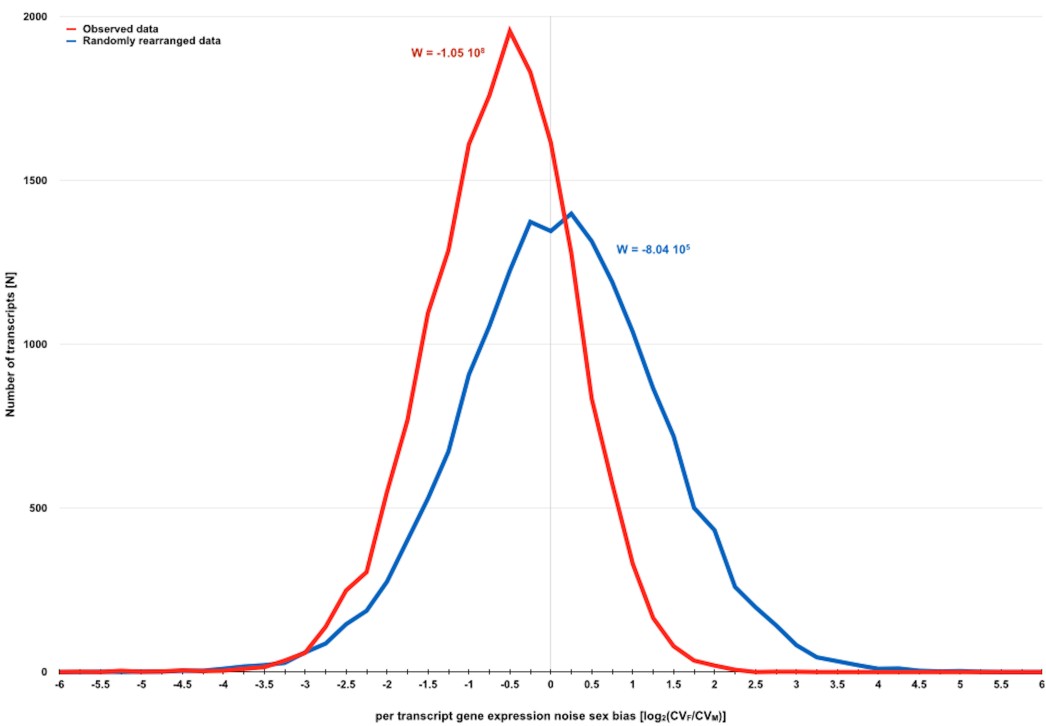

**Figure 2  Direct evidence for the existence of a sexual dimorphism in gene expression noise in _D. melanogaster_.** Distribution of measures for per transcript gene expression noise sex bias before and after randomly rearranging observed gene expression noise measures for _D. melanogaster strains_ with SIM/REV genotype according to the dataset of _Diaz-Castillo, Xia & Ranz (2012)_. Gene expression noise is measured as transcript abundance coefficient of variation in females and males ($CV_F$ and $CV_M$). Sexual dimorphism in gene expression noise for the whole transcriptome before and after randomly rearranging the observed data was measured using W (see main text for further details). The negative skew of the distribution of observed data is consistent with the hypothesis that gene expression is generally noisier in males than in females.

than for analyses of data from multiple strains sharing the same genotype. Since there is a difference in the extent of biological replication between single- and multiple-strain analyses (Table 1), it is possible that non-significant observed W, as it is the case of strain INV2, might be related with power limitations imposed by lower levels of biological replication. In fact, non-significant trends exclusively associated with data encompassing lower levels of biological replication are also noticeable in forthcoming analyses, which stresses the importance of larger biological replication for the study of biological variation of putative stochastic nature.

The study of the association between transcript abundance mean and CV for each sex in the dataset under study also supports the existence of a sexual dimorphism in gene expression noise in _D. melanogaster_, and hints about its potential cause. It is commonly accepted that the variation in gene expression noise is mostly associated with changes in the mean level of gene expression (_Bar-Even et al., 2006_; _Newman et al., 2006_; _Taniguchi et al., 2010_). For those cases in which gene expression noise was modulated by factors other than the variation in the mean level of expression, a weak association between measures for the mean level and stochastic variation in gene expression should be expected. In the dataset

**Table 2** Coefficients of determination ($R^2$) for quadratic regressions between transcript abundance mean and CV for *D. melanogaster* females and males.

| Strains | Females | Males | Females/Males |
|---------|---------|-------|---------------|
| REC | 0.0496 | 0.0294 | 1.6834 |
| INV1 | 0.0444 | 0.0750 | 0.5923 |
| INV2 | 0.1218 | 0.0619 | 1.9677 |
| SIM1 | 0.0918 | 0.0475 | 1.9333 |
| REV1 | 0.0467 | 0.0123 | 3.7853 |
| REV2 | 0.0639 | 0.0719 | 0.8885 |
| INV | 0.1563 | 0.0608 | 2.5707 |
| SIM/REV | 0.1617 | 0.0281 | 5.7585 |

**Notes.**

Transcript abundance data was obtained from *Diaz-Castillo, Xia & Ranz (2012)*.

of *Diaz-Castillo, Xia & Ranz (2012)*, the coefficient of determination for the quadratic regression between transcript abundance mean and CV is considerably larger in females than in males (Table 2), with the exception of INV1 and REV2. These results suggest that the association between the mean level and stochastic variation in gene expression is weaker in males than in females. The male reduction in gene expression noise variation that could be explained solely through changes in the mean level of gene expression is consistent with the possibility that factors specific to males, heterogametic in *Drosophila*, promoted gene expression noise.

## Indirect evidence for the existence of a sexual dimorphism in gene expression noise in *Drosophila*

It could be argued that the genome-wide difference in gene expression noise between *D. melanogaster* females and males reported here is an aberration of a particular genomic background and/or the use of a particular gene expression quantification methodology. For example, the experimental setup that resulted in *Diaz-Castillo, Xia & Ranz*'s (*2012*) dataset did not include technical replicates. Although it seems unlikely that consistent differences in the processing of female and male samples were to fully account for the generalized difference in transcript abundance variation detected in the dataset in question, it is currently not possible to remove the variation from the measures of putative gene expression noise due to technical issues. Other analyses using different *D. melanogaster* strains, other metazoan species, and/or other quantitative methods to measure gene expression are needed to confirm the existence and characterize further the hypothesized sexual dimorphism in gene expression noise in metazoan species.

Notwithstanding these limitations, the existence of a sexual dimorphism in gene expression noise in metazoan species should also be indirectly tested by studying sexual differences for the intrinsic variation in other phenotypic traits. Since the propagation of gene expression noise across biological levels would result in genotype- and environment-independent variation in other phenotypic traits that might permit populations to cope with environmental changes (*Burga & Lehner, 2012*; *Feinberg & Irizarry, 2010*; *Kaern et al.,*

**Table 3** Monte Carlo–Wilcoxon matched-pairs signed-ranks tests for putative stochastic variation in phenotypic traits for *D. melanogaster* females and males.

| Trait | *N* replicates | *N* data pairs | Observed W | Simulated W [5th/95th percentiles] | $P_{upper}/P_{lower}$ |
|---|---|---|---|---|---|
| Startle response | 18–40 | 405 | −13,773 | −7,788/7,639 | 0.9986/0.0014 |
| Starvation resistance | 2–11 | 203 | −3,176 | −2,724/2,780 | 0.9722/0.0280 |
| Chill coma recovery | 2–4 | 174 | −1,395 | −2,201/2,151 | 0.8527/0.1476 |

**Notes.**

Phenotypic traits data was obtained from http://dgrp2.gnets.ncsu.edu (*Mackay et al., 2012*). Observed Wilcoxon W were obtained by subtracting $CV_M$ from $CV_F$ for each element in the dataset for each strain or genotype. CV differences were ranked from lower to higher according to their absolute value. Signs were assigned to ranks according to the sign of CV differences. W were obtained by adding signed ranks for all the elements in the dataset for each strain or genotype. Simulated W were obtained by repeating the same process after randomly rearranging all CV values for each strain or genotype 10,000 times. $P_{upper}$ and $P_{lower}$ values represent the fraction of random simulations with measures larger or equal, and lower or equal, than the observed ones, respectively.

*2005*; *Kilfoil, Lasko & Abouheif, 2009*; *Lehner, 2013*; *Raj & Van Oudenaarden, 2008*; *Raser & O'Shea, 2005*), the existence of a sexual dimorphism in gene expression noise could be indirectly supported by clear sexual differences in the variation of phenotypic traits related to interaction with the environment.

The *D. melanogaster* Genetic Reference Panel (DGRP) is a collection of over 200 inbred lines derived from a single *D. melanogaster* population in Raleigh (North Carolina, USA)(*Huang et al., 2014*; *Mackay et al., 2012*). The generation of these lines is aimed at encompassing standing genetic variation present in the founder population, so their intensive genotypic and phenotypic characterization helped to address the relationship between the variation in genes and phenotypes. Among other assays, females and males from DGRP lines were used to measure their response to three environmental stressors: startle response, starvation, cold (*Mackay et al., 2012*). If the sexual dimorphism in gene expression noise detected in the dataset of *Diaz-Castillo, Xia & Ranz (2012)* was extensive in all *D. melanogaster* populations, and a generalized difference in gene expression noise between sexes resulted in a sexual difference in phenotypic variation, it would be expected that, regardless of genotypic and environmental differences, the response to these stressors were more variable in males than in females for the DGRP lines.

To test this prediction, I performed Monte Carlo–Wilcoxon matched-pairs signed-ranks tests with DGRP startle response, starvation resistance and chill comma recovery data (http://dgrp2.gnets.ncsu.edu) (*Mackay et al., 2012*). I calculated $CV_F$ and $CV_M$ for groups of females and males from the same strain assayed in the same conditions. Although these three traits were originally measured individually, the assortment of assayed individuals was different for each trait. Data arrangement to best accommodate it in pairs of same genotype females and males assayed in the same conditions resulted in differences in the level of biological replication and the number of elements under study for each trait (Table 3). Startle response was assayed using groups of individuals of the same sex per vial for two different dates (*Mackay et al., 2012*). Since environmental conditions between dates might be slightly different, I considered assays for different days separately. I considered individuals within each group as independent biological replicates, and calculated startle

response $CV_F$ and $CV_M$, using measures for individuals of the same strain and sex assayed simultaneously. Starvation resistance was assayed using groups of individuals of the same sex per vial simultaneously (*Mackay et al., 2012*). I considered groups of individuals of the same strain and sex as independent biological replicates, and averaged the starvation response measures for individuals in the same group. I calculated starvation resistance $CV_F$ and $CV_M$, using averaged measures for each biological replicate of the same strain and sex. Chill comma recovery was assayed using the same strain and sex groups of single individual vials assayed simultaneously (*Mackay et al., 2012*). I considered each group of individuals of the same strain and sex as independent biological replicates, and averaged the chill comma recovery measures for individuals in the same group. I calculated chill comma recovery $CV_F$ and $CV_M$, using averaged measures for each biological replicate of the same strain and sex.

For each pair of $CV_F$ and $CV_M$, the latter was subtracted from the former. CV differences were ranked according to their absolute value (from lower to higher), and signs were assigned to each rank upon assertaining the difference between $CV_F$ and $CV_M$. W was calculated for each trait using the observed assortment of CV, and after randomly rearranging all CV data for each trait 10,000 times. As expected, if gene expression noise-dependent phenotypic variation was generally larger in males than in females, observed W were negative for all three traits (Table 3). Moreover, for startle response and starvation resistance, the fraction of simulated W lower than observed W was below the common threshold of significance ($P < 0.05$) (Table 3). Although chill comma recovery observed in W was not significantly different from that in simulated W ($P > 0.05$), it is worth noting that this trait is considered to have the lowest level of biological replication; however, observed W was lower than the majority of simulated W (8,527 out of 10,000 simulated W are larger than the observed one) (Table 3). Further analyses of sexual differences in the intrinsic variation of chill comma recovery with larger biological replication would be needed to decide if this trait shows a similar trend as the other two.

### Indirect evidence for the existence of a sexual dimorphism in gene expression noise in Metazoa

Sexual differences in gene expression noise with effects on phenotypic traits related to interaction with the environment could influence the capacity of each sex to endure environmental changes and, therefore, their spatiotemporal dynamics. For instance, if a larger gene expression noise resulted in a phenotypic variation that facilitated enduring environmental changes, it would be expected that noisier heterogametic individuals were able to better survive drastic environmental changes or occupy broader and/or more diverse geographies than less noisy homogametic individuals.

The movement of individuals away from their birth place, or dispersal, is constrained both by environmental conditions and factors intrinsic to dispersing individuals (*Clobert et al., 2012*). It is known that dispersal is sexually dimorphic in many species (*Clobert et al., 2012*; *Dobson, 2013*; *Greenwood, 1980*). For example, in mammals males tend to disperse more than females, whereas in birds females are the ones who disperse more than the

**Table 4** Monte Carlo simulations for heterogamety-biased dispersal in metazoan species with and without sex-specific heterochromatic chromosomes.

| Species | Observed fraction of heterogamety-biased dispersal | Simulated fraction of heterogamety-biased dispersal [5th–95th percentiles] | $P_{upper}/P_{lower}$ |
|---|---|---|---|
| All | 0.81 | 0.36/0.64 | 0.0005/1.0000 |
| XY + ZW | 0.90 | 0.36/0.65 | 0.0001/1.0000 |
| XO + ZO | 0.20 | 0.20/0.80 | 0.9731/0.1788 |

**Notes.**

Sex-biased dispersal and chromosome system data were obtained from *Spence (1990)*, *Morgan-Richards (1997)*, *Kandul, Lukhtanov & Pierce (2007)*, *Ardila-Garcia & Gregory (2009)*, *Petit & Excoffier (2009)* and *Narita et al. (2011)*. XO and ZO represent species in which heterogametic individuals commonly or completely lack W and Y. The fraction of heterogamety-biased dispersal represents the number of cases in the dataset in which heterogametic individuals tend to disperse more than homogametic individuals. Simulated fractions of heterogamety-biased dispersal were calculated after randomly rearranging chromosome system tags 10,000 times. $P_{upper}$ and $P_{lower}$ values represent the fraction of random simulations with measures larger or equal, and lower or equal than the observed ones, respectively.

males (*Clobert et al., 2012*; *Dobson, 2013*; *Greenwood, 1980*). Since mammal males and bird females are heterogametic, their ability to disperse more than their homogametic relatives would fit with an enhanced ability to endure environmental changes, possibly granted by a generalized larger gene expression noise in heterogametic individuals. The study of karyotypic constraints on sex-biased dispersal lends indirect support to the existence of the sexual dimorphism in gene expression noise in metazoan species promoted by heterogametic-specific factors.

*Petit & Excoffier (2009)* studied the contribution of different factors to interspecific gene flow, using a dataset that included 36 widely distributed metazoan species for which sex-biased dispersal trends and chromosome systems were known. In that dataset, birds, insects, and mammals were represented by 11, 11, and 14 cases, respectively (Table S1). To confirm that sex-biased dispersal could indeed reflect a generalized larger gene expression noise in heterogametic metazoans, I calculated the fraction of cases within the Petit and Excoffier dataset where the heterogametic sex was the one dispersing the most, i.e., the fraction of heterogamety-biased dispersal. To test the significance of observed heterogamety-biased dispersal, I also calculated the fraction of heterogamety-biased dispersal after randomly rearranging the chromosome system tags 10,000 times. As expected, if metazoan heterogametic individuals tended to disperse more than homogametic individuals, then the observed fraction of heterogamety-biased dispersal was significantly larger than expected by chance (Table 4).

Interestingly, the Petit and Excoffier dataset includes five cases in which species commonly or completely lack sex-specific chromosomes (W/Y chromosomes hereinafter) (Table S1) (*Ardila-Garcia & Gregory, 2009*; *Kandul, Lukhtanov & Pierce, 2007*; *Morgan-Richards, 1997*; *Narita et al., 2011*; *Petit & Excoffier, 2009*; *Spence, 1990*). When the fraction of heterogamety-biased dispersal was calculated separately for species in which heterogametic individuals carried or lacked W/Y chromosomes, it was shown that heterogametic individuals significantly dispersed the most only if they carried W/Y chromosomes (Table 4). These results suggest that the presence of W/Y chromosomes

is an even better predictor for dispersal than heterogamety. Considering argued that differences in sex-biased dispersal could be the consequence of sexual differences in the variation in phenotypic traits that influence interaction with the environment and is ultimately caused by a generalized difference in gene expression noise between sexes, $W/Y$ chromosomes-driven dispersal could hint about a potential connection between $W/Y$ chromosomes and an increase in putative gene expression noise-based phenotypic variation. This inference agrees with the possibility that in *D. melanogaster* gene expression noise is promoted by the male-specific factors suggested by the study of *Diaz-Castillo, Xia & Ranz*'s (*2012*) dataset. The connection between $W/Y$ chromosomes and gene expression noise requires further consideration.

## The sexual dimorphism in gene expression noise might depend on *W/Y* chromosomes acting as genomic tuning knobs

In *D. melanogaster*, the $Y$ chromosome has been shown to have a modulator effect on phenotypic variation of stochastic nature and gene expression variation across the genome. On one side, the *D. melanogaster* $Y$ chromosome acts as a suppressor of the phenomenon known as position effect variegation (PEV); namely, the stochastic inactivation of genes when they relocate into or are juxtaposed to regions with highly compacted chromatin, i.e., heterochromatin (*Elgin & Reuter, 2013*; *Gowen & Gay, 1934*). On the other side, variation in the *D. melanogaster* $Y$ chromosome has been shown to have an indirect effect on the expression of multiple genes across the genome, a phenomenon referred to as $Y$-linked regulatory variation (YRV) (*Lemos, Branco & Hartl, 2010*; *Paredes et al., 2011*; *Sackton & Hartl, 2013*; *Zhou et al., 2012*). Although YRV affects genes spread widely in the genome, it tends to be more accentuated for loci located in an environment where gene expression is actively repressed, i.e., heterochromatin and nuclear periphery (*Sackton & Hartl, 2013*). Since the indirect effects of the *D. melanogaster* $Y$ chromosome seem to be heterochromatin-centric, and the *D. melanogaster* $Y$ chromosome encodes for very few genes with functionalities not obviously connected with the regulation of gene expression, it has been argued that the indirect effects of the *D. melanogaster* $Y$ chromosome can depend on it acting as a sink for heterochromatin-forming elements (*Berloco et al., 2014*; *Sackton & Hartl, 2013*; *Zuckerkandl, 1974*). The highly heterochromatic *D. melanogaster* $Y$ chromosome requires such a large amount of the elements needed for heterochromatin formation that these elements would become depleted in heterochromatic loci located in other chromosomes, i.e., non-$Y$ heterochromatic loci. $Y$ chromosome-mediated depletion in heterochromatin-forming elements in non-$Y$ loci will affect their level of chromatin compaction and, therefore, their access to the transcription machinery. Since some of the elements required for heterochromatin formation play also roles in non-heterochromatic loci (*Cryderman et al., 2005*; *Fanti et al., 2008*), the *D. melanogaster* $Y$ chromosome sink effect would be heterochromatin-centric but not heterochromatin-exclusive.

$W/Y$ chromosomes have originated from autosomes independently on several occasions, but they all seem to proceed through a progressive loss of coding elements and enrichment in repetitive DNA (*Bachtrog, 2013*; *Ellegren, 2011*; *Mank, 2012*; *O'Meally et*

*al., 2010*). Except for the chromosomes in early stages of this process, *W/Y* chromosomes tend to be the largest repositories in repetitive DNA in the genome, making them the largest heterochromatic bodies in the nucleus (*Ellegren, 2011*; *Mank, 2012*; *O'Meally et al., 2010*). To test if the putative dependence on *W/Y* chromosomes of the sexual dimorphism in gene expression noise documented here was based on these chromosomes acting as a sink for heterochromatic-forming elements, I returned to the dataset of *Diaz-Castillo, Xia & Ranz (2012)*.

*Filion et al. (2010)* used genome-wide binding patterns of 53 chromatin elements to define five components in the *D. melanogaster* genome, which they symbolized with different colors. Four of these components are related to stable or transitory gene repression. BLUE and GREEN represent known heterochromatin repositories. BLACK and RED are enriched in loci with tissue-restricted gene expression and lamin-binding targets, suggesting they might be located towards the repressive environment of the nuclear periphery. YELLOW is the only component associated with broadly expressed loci. Transcripts in the dataset of Diaz-Castillo and associates were assigned colors if the genomic region where they were encoded according to *D. melanogater* genome annotation were spanned in their entirety by a single-color tract according to Filion and coworkers (*St Pierre et al., 2014*). As a measure for sexual dimorphism in gene expression noise, I calculated W for transcripts in each color per strain and genotype, following the previously explained process. The significance of the sexual dimorphism in gene expression noise for loci in each color was evaluated by recalculating W after randomly rearranging 10,000 times all CV for each strain and genotype.

A common pattern in strains and genotypes found in the dataset is that observed W are negative and considerably smaller for YELLOW transcripts than for transcripts of any other color (Fig. 3, and Table 5). Such a pattern suggests that male-biased gene expression noise is less extreme for loci within repressive areas of the genome, which is consistent with what would be expected if the *D. melanogaster Y* chromosome sink effect altered chromatin compaction across the genome and, consequently, gene expression noise. Since much of the material required for heterochromatin formation in males would be diverted towards the *Y* chromosome, non-*Y* heterochromatic loci would be expected to be more compacted in females than in males (Fig. 4). Higher chromatin compaction structures have been associated with an increase in gene expression noise due to slower gene expression dynamics (*Kaern et al., 2005*; *Raj & Van Oudenaarden, 2008*; *Raser & O'Shea, 2005*). Thus, non-*Y* heterochromatic loci would be expected to be considerably more noisy in females and, therefore, show less extreme sexual dimorphism in gene expression noise.

The analysis of the sexual dimorphism in gene expression noise for genes in different compartments also showed that, with the exception of INV2 and REV2, observed W were negative and smaller than any simulated W for each chromatin color (Table 5). These results suggest that gene expression is generally noisier in males than in females, regardless of chromatin compaction and subnuclear localization. Although the deployment of heterochromatin-forming elements in the presence/absence of *W/Y* chromosomes might cause differences in chromatin compaction and gene expression noise across genomes, on

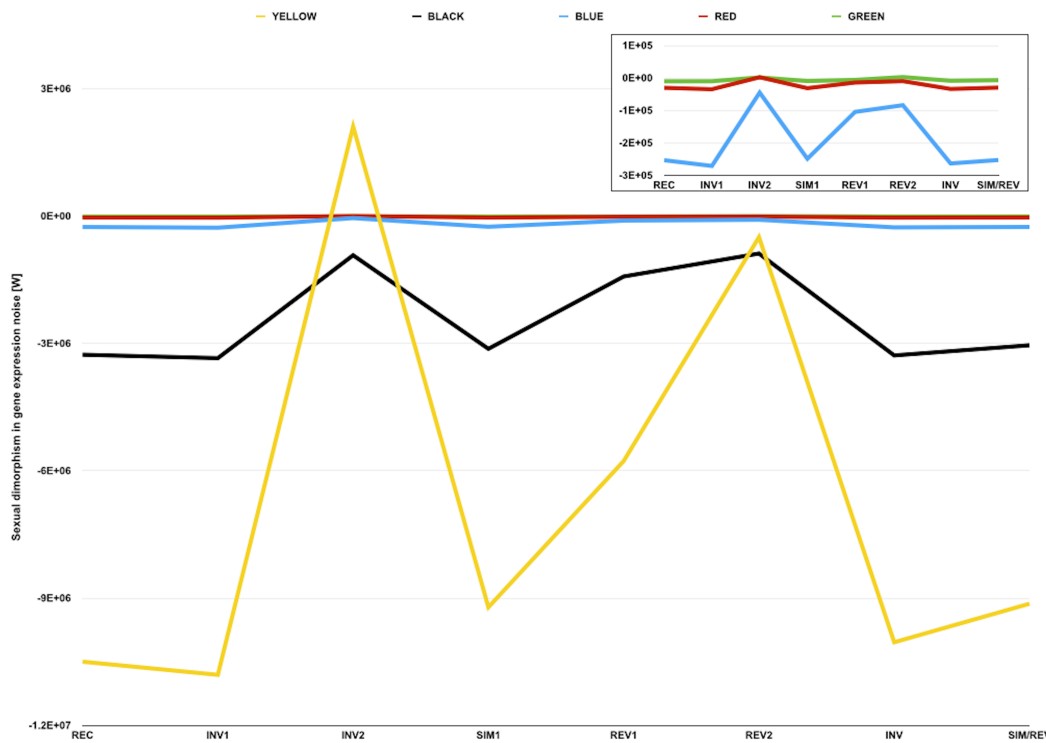

**Figure 3 Sexual dimorphism in gene expression noise for genes in five different compartments of the *D. melanogaster* genome.** The graph represents measures of sexual dimorphism in gene expression noise for loci with different chromatin structure and/or subnuclear localization according to *Filion et al. (2010)*, in *D. melanogaster* strains and genotypes represented in the dataset of *Diaz-Castillo, Xia & Ranz (2012)*. BLUE and GREEN components represent known heterochromatin repositories (*Filion et al., 2010*). BLACK and RED are enriched in loci with tissue-restricted gene expression and lamin-binding targets, suggesting they might be located towards the repressive environment of the nuclear periphery (*Filion et al., 2010*). YELLOW is the only component associated with broadly expressed loci (*Filion et al., 2010*). Sexual dimorphism in gene expression noise was measured using W (see main text for further details). The area of the graph representing W for BLUE, RED, and GREEN loci is amplified in the inner box.

its own a heterochromatin sink effect common to all $W/Y$ chromosomes could not fully explain the existence of the putative sexual dimorphism in the gene expression noise in metazoan species.

*King, Soller & Kashi (1997)* proposed the concept of "genetic tuning knobs" to refer to the indirect effect that the variation in repetitive DNA has on genetic elements they belong to or with which they are intimately associated. Small, reversible, and very frequent copy number variation in repetitive DNA would have a fine-tuning modifier effect on coding units and/or non-coding motives with regulatory attributes (*Gemayel et al., 2010*; *Kashi & King, 2006*; *King, Soller & Kashi, 1997*). If repetitive DNA motives in $W/Y$ chromosome were susceptible to frequent variation in copy number, it could be expected that the fraction of repetitive DNA in $W/Y$ chromosomes was a very variable trait. Since the heterochromatic nature of $W/Y$ chromosomes depends on their content in repetitive DNA, it could be inferred that $W/Y$ chromosomes with slightly different fractions of repetitive DNA would withdraw slightly different amounts from the limiting

**Table 5 Monte Carlo–Wilcoxon matched-pairs signed-ranks tests for putative stochastic variation in transcript abundance in five different chromatin compartments of *D. melanogaster* females and males.**

| Strains | Chromatin color | N replicates | N data pairs | Observed W | Simulated W [5th/95th percentiles] | $P_{upper}/P_{lower}$ |
|---|---|---|---|---|---|---|
| REC | YELLOW | 3 | 4,907 | $-1.05 \times 10^7$ | $-3.30 \times 10^5/3.20 \times 10^5$ | 1.0000/<0.0001 |
| REC | BLACK | 3 | 2,721 | $-3.26 \times 10^6$ | $-1.36 \times 10^5/1.34 \times 10^5$ | 1.0000/<0.0001 |
| REC | BLUE | 3 | 775 | $-2.53 \times 10^5$ | $-2.06 \times 10^4/2.01 \times 10^4$ | 1.0000/<0.0001 |
| REC | RED | 3 | 266 | $-3.00 \times 10^4$ | $-4.19 \times 10^3/4.06 \times 10^3$ | 1.0000/<0.0001 |
| REC | GREEN | 3 | 147 | $-1.00 \times 10^4$ | $-1.70 \times 10^3/1.67 \times 10^3$ | 1.0000/<0.0001 |
| INV1 | YELLOW | 3 | 4,907 | $-1.08 \times 10^7$ | $-3.32 \times 10^5/3.22 \times 10^5$ | 1.0000/<0.0001 |
| INV1 | BLACK | 3 | 2,721 | $-3.34 \times 10^6$ | $-1.36 \times 10^5/1.36 \times 10^5$ | 1.0000/<0.0001 |
| INV1 | BLUE | 3 | 775 | $-2.71 \times 10^5$ | $-2.06 \times 10^4/2.10 \times 10^4$ | 1.0000/<0.0001 |
| INV1 | RED | 3 | 266 | $-3.44 \times 10^4$ | $-4.14 \times 10^3/4.17 \times 10^3$ | 1.0000/<0.0001 |
| INV1 | GREEN | 3 | 147 | $-9.28 \times 10^3$ | $-1.71 \times 10^3/1.68 \times 10^3$ | 1.0000/<0.0001 |
| INV2 | YELLOW | 3 | 4,907 | $2.11 \times 10^6$ | $-3.29 \times 10^5/3.27 \times 10^5$ | <0.0001/1.0000 |
| INV2 | BLACK | 3 | 2,721 | $-9.20 \times 10^5$ | $-1.37 \times 10^5/1.36 \times 10^5$ | 1.0000/<0.0001 |
| INV2 | BLUE | 3 | 775 | $-4.46 \times 10^4$ | $-2.11 \times 10^4/2.03 \times 10^4$ | 0.9999/0.0001 |
| INV2 | RED | 3 | 266 | $2.56 \times 10^3$ | $-4.19 \times 10^3/4.15 \times 10^3$ | 0.1497/0.8505 |
| INV2 | GREEN | 3 | 147 | $2.17 \times 10^3$ | $-1.70 \times 10^3/1.66 \times 10^3$ | 0.0149/0.9851 |
| SIM1 | YELLOW | 3 | 4,907 | $-9.21 \times 10^6$ | $-3.33 \times 10^5/3.28 \times 10^5$ | 1.0000/<0.0001 |
| SIM1 | BLACK | 3 | 2,721 | $-3.12 \times 10^6$ | $-1.34 \times 10^5/1.37 \times 10^5$ | 1.0000/<0.0001 |
| SIM1 | BLUE | 3 | 775 | $-2.48 \times 10^5$ | $-2.02 \times 10^4/2.07 \times 10^4$ | 1.0000/<0.0001 |
| SIM1 | RED | 3 | 266 | $-3.11 \times 10^4$ | $-4.17 \times 10^3/4.12 \times 10^3$ | 1.0000/<0.0001 |
| SIM1 | GREEN | 3 | 147 | $-9.26 \times 10^3$ | $-1.70 \times 10^3/1.72 \times 10^3$ | 1.0000/<0.0001 |
| REV1 | YELLOW | 3 | 4,907 | $-5.77 \times 10^6$ | $-3.25 \times 10^5/3.21 \times 10^5$ | 1.0000/<0.0001 |
| REV1 | BLACK | 3 | 2,721 | $-1.42 \times 10^6$ | $-1.34 \times 10^5/1.37 \times 10^5$ | 1.0000/<0.0001 |
| REV1 | BLUE | 3 | 775 | $-1.04 \times 10^5$ | $-2.05 \times 10^4/2.03 \times 10^4$ | 1.0000/<0.0001 |
| REV1 | RED | 3 | 266 | $-1.37 \times 10^4$ | $-4.11 \times 10^3/4.09 \times 10^3$ | 1.0000/<0.0001 |
| REV1 | GREEN | 3 | 147 | $-5.61 \times 10^3$ | $-1.69 \times 10^3/1.71 \times 10^3$ | 1.0000/<0.0001 |
| REV2 | YELLOW | 3 | 4,907 | $-4.94 \times 10^5$ | $-3.27 \times 10^5/3.29 \times 10^5$ | 0.9930/0.0070 |
| REV2 | BLACK | 3 | 2,721 | $-8.79 \times 10^5$ | $-1.34 \times 10^5/1.33 \times 10^5$ | 1.0000/<0.0001 |
| REV2 | BLUE | 3 | 775 | $-8.35 \times 10^4$ | $-2.06 \times 10^4/2.02 \times 10^4$ | 1.0000/<0.0001 |
| REV2 | RED | 3 | 266 | $-9.38 \times 10^3$ | $-4.04 \times 10^3/4.15 \times 10^3$ | 0.9998/0.0002 |
| REV2 | GREEN | 3 | 147 | $3.15 \times 10^3$ | $-1.66 \times 10^3/1.68 \times 10^3$ | 0.0010/0.9990 |
| INV | YELLOW | 6 | 4,907 | $-1.00 \times 10^7$ | $-3.29 \times 10^5/3.22 \times 10^5$ | 1.0000/<0.0001 |
| INV | BLACK | 6 | 2,721 | $-3.28 \times 10^6$ | $-1.35 \times 10^5/1.38 \times 10^5$ | 1.0000/<0.0001 |
| INV | BLUE | 6 | 775 | $-2.63 \times 10^5$ | $-2.06 \times 10^4/2.07 \times 10^4$ | 1.0000/<0.0001 |
| INV | RED | 6 | 266 | $-3.35 \times 10^4$ | $-4.11 \times 10^3/4.09 \times 10^3$ | 1.0000/<0.0001 |
| INV | GREEN | 6 | 147 | $-8.39 \times 10^3$ | $-1.68 \times 10^3/1.70 \times 10^3$ | 1.0000/<0.0001 |
| SIM/REV | YELLOW | 9 | 4,907 | $-9.13 \times 10^6$ | $-3.21 \times 10^5/3.30 \times 10^5$ | 1.0000/<0.0001 |
| SIM/REV | BLACK | 9 | 2,721 | $-3.04 \times 10^6$ | $-1.33 \times 10^5/1.33 \times 10^5$ | 1.0000/<0.0001 |
| SIM/REV | BLUE | 9 | 775 | $-2.52 \times 10^5$ | $-2.07 \times 10^4/2.07 \times 10^4$ | 1.0000/<0.0001 |
| SIM/REV | RED | 9 | 266 | $-2.94 \times 10^4$ | $-4.16 \times 10^3/4.15 \times 10^3$ | 1.0000/<0.0001 |
| SIM/REV | GREEN | 9 | 147 | $-6.51 \times 10^3$ | $-1.69 \times 10^3/1.73 \times 10^3$ | 1.0000/<0.0001 |

**Notes.**

Transcript abundance data was obtained from *Diaz-Castillo, Xia & Ranz (2012)*. Coordinates of chromatin components symbolized with colors were obtained from *Filion et al. (2010)*. Observed Wilcoxon W were obtained by subtracting $CV_M$ from $CV_F$ for each element in every chromatin compartment for each strain or genotype. CV differences were ranked from lower to higher according to their absolute value. Signs were assigned to ranks according to the sign of CV differences. W were obtained by adding signed ranks for all the elements with the same color for each strain or genotype. Simulated W were obtained by repeating the same process after randomly rearranging all CV values for each strain or genotype 10,000 times. $P_{upper}$ and $P_{lower}$ values represent the fraction of random simulations with measures larger or equal, and lower or equal than the observed ones, respectively.

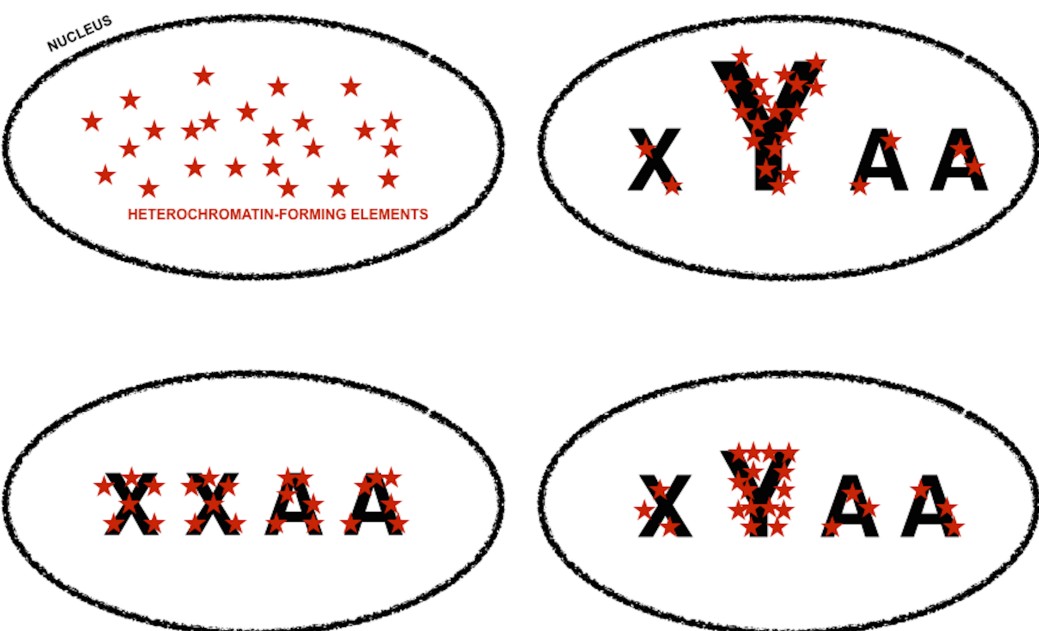

**Figure 4 Genomic tuning knob-sink effect of *Y* chromosomes.** Model for the assortment of heterochromatin-forming elements in homogametic and heterogametic nuclei, under the assumption that heterochromatic-forming elements are found in similar and limiting amounts in all nuclei. Font size is used to symbolize differences in the repetitive DNA content of sexual chromosomes, i.e., *X* and *Y*, and autosomes (*A*). The fraction of heterochromatin-forming elements deployed in *Y* chromosomes can vary depending on their content in repetitive DNA. The assortment of heterochromatin-forming elements in non-*Y* loci would be very different if the nuclei carry or lack *Y* chromosomes, i.e., *XYAA* or *XXAA*, and slightly different if the *Y* chromosomes have different amounts of repetitive DNA. Slight differences in chromatin compaction across the genome ultimately based in small differences in *Y* chromosome repetitive DNA will cause an increase in gene expression noise across the genome, whereas higher level of chromatin compaction in non-*Y* heterochromatic loci in *XXAA* nuclei will be translated into less extreme male-biased gene expression noise in these loci (Fig. 3, and main text).

pool of heterochromatin-forming elements (Fig. 4). The variation in the amount of heterochromatin-forming elements deployed in *W/Y* chromosomes with slightly different fractions of repetitive DNA would be translated into a variation in chromatin compaction in genes across the genome and, therefore, in their access to the transcription machinery (Fig. 4). Thus, a small variation in the fraction of repetitive DNA in *W/Y* chromosomes could ultimately cause an increase in gene expression noise across the genome of heterogametic individuals. In other words, the sexual dimorphism in gene expression noise in metazoan species could be the consequence of *W/Y* chromosomes acting as tuning knobs at a genomic scale. Following with the musical theme, *W/Y* chromosomes could be compared to the pulley-and-lever system of single-stringed whamolas. Handling of a whamola lever modifies the tension of the string causing a variation in noise when played. Similarly, variation in *W/Y* chromosomes repetitive DNA would modify chromatin compaction across the genome and, with it, access to the transcription machinery.

The characterization of genome regions enriched in repetitive DNA is one of the unresolved challenges in genomics (*Chain et al., 2009*; *Treangen & Salzberg, 2012*). In

fact, genomic regions enriched in repetitive DNA, such as $W/Y$ chromosomes, are remarkably misrepresented in genome assemblies. The confirmation of the role of $W/Y$ chromosomes as genomic tuning knobs, and its importance for the sexual dimorphism in gene expression noise documented here, are currently hard to address empirically. Despite current technical limitations, in *D. melanogaster* there is indirect evidence consistent with the possibility that the $Y$ chromosome might act as a genomic tuning knob. On one side, large and subtley induced changes in the fraction of repetitive DNA in the *D. melanogaster* $Y$ chromosome have effects similar to PEV modulation and YRV (*Berloco et al., 2014*; *Dimitri & Pisano, 1989*; *Paredes et al., 2011*). Also, the fraction of repetitive DNA in $W/Y$ chromosomes has been shown to be very variable between closely-related species and natural populations of a single species (*Halfer, 1981*; *Hughes & Rozen, 2012*; *Lyckegaard & Clark, 1989*; *Lyckegaard & Clark, 1991*; *Nova et al., 2002*; *Paredes et al., 2011*; *Repping et al., 2006*; *Sahara, Yoshido & Traut, 2012*; *Singh, Purdom & Jones, 1980*). In *D. melanogaster*, repetitive DNA derivatives thought to mediate its copy number plasticity have been detected for the $Y$ chromosome in individuals of a single strain (*Cohen et al., 2005*; *Cohen & Segal, 2009*), which is consistent with the possibility that *D. melanogaster* $Y$ chromosome repetitive DNA variation can be spontaneous and frequent. Thus, at least for the *D. melanogaster* $Y$ chromosome, independent lines of evidence exist which are consistent with a frequent variation in its repetitive DNA, and that this type of variation has an effect on gene expression across the genome, as would be expected if this chromosome acted as a genomic tuning knob. Further research is required to confirm if $W/Y$ chromosomes acting as genomic tuning knobs cause a sexual dimorphism in gene expression noise in *D. melanogaster* and other metazoan species.

## The sexual dimorphism in gene expression noise might depend on nuclear dynamics in gametogenesis and early embryogenesis

The use of the $W/Y$ chromosomes genomic tuning knob-sink effect model to explain sexual differences in gene expression noise in metazoan species is contingent not only upon the spontaneous and frequent variation in the repetitive DNA of $W/Y$ chromosomes, but also upon heterochromatin-forming elements in being roughly the same limiting amounts in the nuclei of both sexes (Fig. 4). There is only one moment common to all metazoans in which the pool of heterochromatin-forming elements might be in comparable limiting amounts regardless of chromosome composition. From oocyte fertilization until the zygotic genome gets fully activated, all new chromatin is formed at the expense of maternally-deposited material (*Banaszynski, Allis & Lewis, 2010*; *Baroux et al., 2008*; *Tadros & Lipshitz, 2009*). Furthermore, sperm chromosomes enter the oocyte in a state of extreme compaction, thanks to the almost complete substitution of histones by protamine-like proteins along spermatogenesis (*Banaszynski, Allis & Lewis, 2010*; *Tadros & Lipshitz, 2009*). During the transformation of sperm nuclei into paternal pronuclei, protamine-like proteins are substituted by histones; a process so intensive that it will take a very large amount of the already limiting maternally-deposited material (*Banaszynski, Allis & Lewis, 2010*; *Tadros & Lipshitz, 2009*). In this context, the deployment

of heterochromatin-forming elements during the first zygotic nuclear divisions might not run the same course in the presence or absence of $W/Y$ chromosomes, or when the amount of repetitive DNA in these chromosomes is different (Fig. 4). It could be speculated that the sexual dimorphism in gene expression noise in metazoan species ultimately depended on $W/Y$ chromosomes acting as genomic tuning knobs-sinks very early in embryogenesis.

Interestingly, evidence exists in *Drosophila* and Lepidoptera for the preferential use of DNA break repair strategies that can cause repetitive DNA variation in the germ line of heterogametic individuals (*Cohen & Segal, 2009*; *Díaz-Castillo, 2013*; *Diaz-Castillo & Ranz, 2012*; *Lieber, 2010*; *Peng & Karpen, 2007*; *Preston, Flores & Engels, 2006*; *Suzuki et al., 2009*). Also, silencing patterns of PEV reporters in *Drosophila*, when maternally- or paternally-inherited, are consistent with heterochromatin in later stages of development dependent on the assortment of heterochromatic-forming elements very early in embryogenesis (*Golic, Golic & Pimpinelli, 1998*; *Maggert & Golic, 2002*). Both instances of the intrinsic spontaneous variation in repetitive DNA for $W/Y$ chromosomes occurred along gametogenesis, and the preservation along the development of early embryogenesis chromatin compaction genomic patterns are required for W/Y chromosomes which act as genomic tuning knobs-sinks in very early embryogenesis to be the base for the sexual dimorphism in gene expression noise in metazoan species. Further research is required to confirm that nuclear dynamics along gametogenesis and early embryogenesis can indeed have such an effect on the intrinsic variability in gene expression.

## CONCLUSIONS

In this article I hypothesized that gene expression might be generally noisier for heterogametic individuals than for homogametic individuals in metazoan species. I presented direct evidence of the hypothesized sexual dimorphism by studying a *D. melanogaster* dataset where transcript variation might be mostly of stochastic and sexual nature. Also, by taking into consideration the direct and capacitance-mediated indirect contribution of gene expression noise to phenotypic variation, I predicted that phenotypic traits related to interaction with the environment should be more variable for heterogametic individuals than for homogametic individuals. Evidence of the putative gene expression noise-mediated dimorphism in phenotypic traits were found by studying genotype- and environment-independent variation of the response to three stressors in *D. melanogaster*, and the sex-biased dispersal in metazoan species. Further analyses might speculate that the sexual dimorphism in gene expression noise might be dependent on sex-specific chromosomes acting as genomic tuning knobs very early in embryogenesis. The intrinsically frequent variation in repetitive DNA for $W/Y$ chromosomes might cause variation in chromatin compaction across the genome, which ultimately is transformed in an increase of gene expression noise. The genomic tuning knob-sink model for the origin of the sexual dimorphism in gene expression noise, which subsequently contributes to sexual differences for the variation of other phenotypic traits, illustrates the difficulty of understanding the connection between genotypes and phenotypes. The $W/Y$ chromosomes genomic

tuning knob-sink effect could be described as epigenetic (i.e., depending on chromatin compaction variation) but also as stochastic and genetic (i.e., depending on spontaneous variation in repetitive DNA). Notwithstanding semantic tribulations, the possibility that chromosomes with very limited coding potential (i.e., $W/Y$ chromosomes) can have genome-wide effects such as PEV modification, YRV or gene expression noise modulation says much about the potential importance that nuclear dynamics might have for the phenotypic expression of genotypes.

## ACKNOWLEDGEMENTS

The author expresses his deepest gratitude to Raquel Chamorro-García for valuable comments during the preparation of this article, and thanks to Richard Jorgensen, Alexander de Luna, and one more anonymous reviewer for their constructive criticism.

### Funding

The author declares there was no funding for this work.

### Competing Interests

The author declares there are no competing interests.

### Author Contributions

- Carlos Díaz-Castillo conceived and designed the experiments, performed the experiments, analyzed the data, contributed reagents/materials/analysis tools, wrote the paper, prepared figures and/or tables, reviewed drafts of the paper.

### Supplemental Information

Supplemental information for this article can be found online at http://dx.doi.org/10.7717/peerj.750#supplemental-information.

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
