# Peer review of "Evidence for a sexual dimorphism in gene expression noise in metazoan species"

_PeerJ, doi:10.7717/peerj.750_

## Round 0.1 · original submission · Major Revisions

Based on the reviews, I think there are two principal concerns that are not minor. One is that the presentation needs to be made more accessible. I recommend getting the help of a non-expert in the area to make sure that the article is accessible not only to those who know this area in depth, but also to other biologists. The second is that the role of technical variation needs to be addressed in a reanalysis of the data, perhaps even by generating more experimental replicates.
Reviewer 2 provides additional comments that should also be addressed in any revision.

Reviewer 1 ·

Basic reporting

This reports an interesting finding and interpretation.
It was difficult to wade through because the writing needs to be clarified and made more accessible.
Other than that I think it should be accepted.

Experimental design

it's fine

Validity of the findings

fine

Additional comments

nice work, very interesting

·

Basic reporting

No comments

Experimental design

No comments

Validity of the findings

No comments

Additional comments

In this manuscript, Carlos Diaz-Castillo presents evidence of dimorphism in gene-expression noise in animals. That is, variation in gene expression levels is different between organisms of different sex. Public data sets were analyzed, showing that there is statistically significant difference between: 1) CV of gene expression between male and female Drosophila flies that are genetically identical; 2) CV of gene expression between males and females under different environmental conditions; 3) Dispersal of individuals in different species from their birth place and the presence/absence of sex-determining chromosomes; and 4) CV of gene expression of Drosophila genes located in five different chromatin states. Based on these analyses, evidence of a correlation between homogametic and heterogametic genotypes and sex-dependent variability of gene expression is provided, along with its possible functional implications under different environments or phenotypes. Moreover, it is proposed that at least part of the gene-expression noise in heterogametic organisms could be explained by how the heterochromatin-forming elements are distributed in sexual chromosomes.
The data sets used are relevant and carefully selected, and statistical analyses are technically sound. I just have two minor concerns and a number of suggestions:
1. At least part of the gene-expression noise is inevitably due to technical variation, and this is not taken into account or discussed as part of the analyses. While the author refers to "putative stochastic variation", it would be useful to be explicit about the components of the value that is being calculated. I wonder if some of the conclusions of this paper could change after decomposing the variation parameter.
2. It is hypothesized that heterochromatin-forming elements redistribute in different ways depending on the presence/absence of sexual chromosomes, leading to the intriguing proposal that different numbers of repetitive regions could modulate biological noise. If I understand correctly, it is being assumed that there is no regulation of the amount of elements depending on sexual chromosomes. Is there data available showing that males and females indeed have the same amount of heterochromatin-forming elements? Would this hypothesis be compatible for both W/Z and X/Y systems?
Suggestions:
* In its current form, it is difficult for the non-specialist to get the main message from the abstract. I suggest to cut down on background section of the abstract and providing more detailed information of what was done and what the main findings are.
* When referring to males and females with the same genotype and conditions, it would be worth mentioning that this is not the case for the sex chromosomes.
*It would be useful to describe in the text which species are included in the dataset of Petit and Excoffier (analysis in Table 4).
*Given that Table 1 shows the main result of this work, I feel that a plot with Cvm against CVf distributions could be more striking and informative.
* Figure 1 could be reorganized to avoid blank plots, or simply delete blank axes
* In Figure 2, it would help if the figure showed which type of expression pattern is represented by each color (as stated in text).
* Figure 3 would be easier to follow if divided in four panels that could be described one at the time in the caption.

---

## Round 0.2 · accepted · Accept

Many thanks for your substantive revisions. Looks great.